# Inferencing Bulk Tumor and Single-Cell Multi-Omics Regulatory Networks for Discovery of Biomarkers and Therapeutic Targets

**DOI:** 10.3390/cells12010101

**Published:** 2022-12-26

**Authors:** Qing Ye, Nancy Lan Guo

**Affiliations:** 1West Virginia University Cancer Institute, Morgantown, WV 26506, USA; 2Lane Department of Computer Science and Electrical Engineering, West Virginia University, Morgantown, WV 26506, USA; 3Department of Occupational and Environmental Health Sciences, School of Public Health, West Virginia University, Morgantown, WV 26506, USA

**Keywords:** biomarkers, therapeutic targets, multi-omics regulatory networks, single cells, Prediction Logic Boolean Implication Networks (PLBINs), network centrality, electronic medical records (EMRs), SEER-Medicare

## Abstract

There are insufficient accurate biomarkers and effective therapeutic targets in current cancer treatment. Multi-omics regulatory networks in patient bulk tumors and single cells can shed light on molecular disease mechanisms. Integration of multi-omics data with large-scale patient electronic medical records (EMRs) can lead to the discovery of biomarkers and therapeutic targets. In this review, multi-omics data harmonization methods were introduced, and common approaches to molecular network inference were summarized. Our Prediction Logic Boolean Implication Networks (PLBINs) have advantages over other methods in constructing genome-scale multi-omics networks in bulk tumors and single cells in terms of computational efficiency, scalability, and accuracy. Based on the constructed multi-modal regulatory networks, graph theory network centrality metrics can be used in the prioritization of candidates for discovering biomarkers and therapeutic targets. Our approach to integrating multi-omics profiles in a patient cohort with large-scale patient EMRs such as the SEER-Medicare cancer registry combined with extensive external validation can identify potential biomarkers applicable in large patient populations. These methodologies form a conceptually innovative framework to analyze various available information from research laboratories and healthcare systems, accelerating the discovery of biomarkers and therapeutic targets to ultimately improve cancer patient survival outcomes.

## 1. Introduction

Despite decades of efforts in cancer research, cancer ranks as the top cause of death and shortened life expectancy in every country in the world [1]. In 2040, the global cancer burden is estimated to increase by 47% from 2020, reaching 28.4 million cases [1]. The Cancer Moonshot project was launched in 2016 to accelerate scientific discovery, foster collaboration, and improve data sharing in cancer research [2]. The current unmet clinical needs in cancer treatment include a lack of biomarkers for precise assessment of cancer risk, tumor progression, recurrence, and treatment response in individual patients. More effective therapeutic targets are needed to improve patient survival outcomes. 

The advent of high-throughput sequencing technology has led to the discovery of abnormal genomic variants in cancer patients as novel therapeutic targets, such as the EML4-ALK fusion gene in non-small-cell lung cancer (NSCLC) [3]. In addition, the blockade of immune checkpoint inhibitors (ICIs), including PD1, PDL1, and CTLA4, has improved cancer patient survival outcomes [4,5,6,7,8,9,10]. However, there are currently no established predictive biomarkers in immunotherapy, as PDL1 and tumor mutational burdens are not proven indicators [11]. Systematic disease mechanisms underlying cancer remain illusive.

The tumor immune microenvironment is a multidimensional system of immune cells, stromal cells, and host factors. Complex and interweaving signaling pathways and networks of genes and proteins function in these various cell types [12]. In the tumor microenvironment, the presence of tertiary lymphoid structures (TLSs) is linked to good cancer prognosis [13]. TLSs comprise B cells and adjacent clusters of dendritic cells and T cells [14]. TLSs and tumor-infiltrating B cells improve ICIs responses in cancer immunotherapy [15,16,17,18]. Recent studies suggest an essential role of B cells in antitumor immunity, including the determination of protective T cell responses in cancer patients [19,20,21]. T-cell dysfunction and therapy have been established for cancer treatment [22,23,24,25,26,27,28]. However, B-cell biology and therapeutic potential have not been substantiated [29,30,31]. The emerging single-cell sequencing technique is an effective method to better understand disease mechanisms and develop novel therapeutic interventions. 

Genes and proteins form complex gene regulatory networks (GRN) in living organisms [32,33]. Perturbed gene regulation is closely related to disease and its revelation is important for developing intervention strategies [34,35,36,37]. Molecular network analysis is crucial to decipher cancer mechanisms and advance precision oncology [38]. Artificial intelligence (AI) methods are needed to reveal essential GRNs and essential hub genes at multiple regulatory levels by analyzing emerging multi-modal data in patient bulk tumors and single cells for the discovery of biomarkers and therapeutic targets. 

To achieve optimal treatment selection in individual patients, it is essential to integrate patient multi-omic biomarkers with clinical, pathological, demographic, and comorbid factors using electronic medical records (EMRs) [39]. Retrospective analysis of EMRs has led to the discovery of new and repositioning drugs [40,41,42]. 

This review is focused on multi-omics data processing and integration in Section 2, common systems biology software and data resources in Section 3, molecular network inference methods in Section 4, hub genes in tumorigenesis, proliferation, and patient survival in Section 5, and integration of multi-omics data with EMRs in Section 6. Finally, we provide recommendations for bulk tumor and single-cell multi-omics network analysis for the discovery of biomarkers and therapeutic targets in Section 7. 

## 2. Bulk Tumor and Single-Cell Multi-Omics Data Analysis 

### 2.1. Multi-Omics Data Processing and Integration

With the rapid development of high-throughput technology, genomic, transcriptomic, proteomic, and metabolomic profiles have provided ample sources of information for researchers to understand molecular disease mechanisms. Nevertheless, data generated from various commercially available platforms and customized arrays pose tremendous challenges for processing, analysis, and integration. The Genome Analysis Toolkit (GATK) is the industry standard for processing multi-omics data in bulk tumors and single cells, including identifying single nucleotides (SNPs) and indels, somatic short variants, copy number variations (CNV), and structural variations (SV) in germline DNA and RNAseq data [43]. In addition to data generated from current sequencing technology, a huge amount of high-throughput data was generated from legacy DNA microarrays. A research group from the FDA reported that biomarkers and predictive models derived from legacy microarray data can accurately predict phenotypes in samples profiled with RNA sequencing, whereas RNA-seq-based models are less accurate in predicting microarray data [44]. This section provides a brief overview of some software packages and methods used for bulk tumor multi-omics data processing and integration. 

#### 2.1.1. Copy Number Variation

Copy number variation (CNV) is a structural variation that is either a duplication or deletion event affecting a large number of base pairs. Deletions, amplifications, gains, and losses collectively termed CNVs, are found in all humans and other mammals [45]. The number of CNVs can make up as much as 5–15% of the human genome [46]. CNVs are a significant source of genomic diversity and driver of somatic and hereditary human diseases including cancer. However, compared to single-nucleotide variations (SNVs), CNVs are still under-investigated, despite their evolutionary significance and clinical relevance. This is a consequence of the inherent challenges in identifying CNVs in diverse populations of cells at low-to-intermediate frequencies [47]. Using a recent method of a fluorescent gene functioning as a single-cell CNV reporter, CNVs are found to occur frequently and undergo selection with predictable dynamics across independently evolving replicate populations [47]. CNVs have been applied in the molecular diagnosis of many diseases and non-invasive prenatal care. Nevertheless, CNVs have not reached their full potential as emerging biomarkers. Cancer immunotherapy targets, including *PD1, PDL1*, *CD27*, and *CD20* have more CNVs than SNVs in NSCLC tumors in The Cancer Genome Atlas (TCGA) [48]. Tumor mutation burdens are used in cancer management, but not CNVs. The screening, diagnosis, prognosis, and monitoring of several illnesses, including cancer and cardiovascular disease, are likely to be significantly impacted by CNVs [49]. 

Genomic alterations in DNA might interfere with the normal function of the genes. The genomic instability and structural dynamics of cancer cells require that CNV data be examined to discover the underlying associations between CNVs, gene/protein expression, and functional aberrations. Different platforms were used to profile genome-scale CNVs, including high-resolution SNP arrays (GeneChip Mapping 250K-Nsp array, Affymetrix), whole-genome tiling path aCGH (BCCRC whole genome tilling path array v2), and whole exome sequencing (SOLiD 5500xl) [50]. Various CNV data processing methods were developed as described below.

PennCNV-Affy [51], a Perl/C-based software tool, is the most commonly used method for CNV calling for data produced with SNP genotyping arrays. The first step is to process the raw CEL files and generate the signal intensity data. The second step is to split the signal file generated from step 1 into individual files. After the file splitting is completed, CNV calls will be generated by PennCNV. The output provides information on the CN state for each SNP probe. Normally, a CN < 2 indicates a deletion in copy number, and a CN > 2 indicates a duplication. For the SNP probes located within the same gene, the probe with the maximum intensity is used to represent the CN state for the gene.

Bioconductor packages CGHbase [52] and CGHcall [53] are often used to call the CNV in the aCGH data. The log_2_ normalized ratios of Cy3/Cy5 are used as inputs. In CGHcall, the number of output classes can be selected among 3 classes (loss, normal, gain), 4 classes (loss, normal, gain, amplification), or 5 classes (double deletion, loss, normal, gain, amplification). 

GISTIC2.0 is a pipeline used to find genes targeted by somatic copy-number alterations (SCNAs) in human cancers [54]. GISTIC2.0 uses an iterative optimization algorithm to deconstruct each segmented copy-number profile into its most likely set of SCNAs. Compared with other methods, GISTIC2.0 is advantageous in separating arm-level and focal SCNAs explicitly by length.

CNV data generated by various platforms provide the corresponding chromosome location of each SNP. To harmonize the CNV data from various platforms, we can convert the genome assembly version from earlier versions, such as hg17, to hg38 by using the Python package *CruzDB*, a fast and intuitive tool for the UCSC genome browser [55]. Using the latest reference genome is an important step to ensure compatibility in the CNV data integration.

#### 2.1.2. Categorization of Gene Regulation 

Cancer is caused by dysregulated tumor suppressor genes or oncogenes. Due to genetic mutations or alterations in gene regulation, such genes are switched on or off and are expressed at abnormally high or low levels in tumor initiation and progression. It is important to define the up-regulation, normal, and down-regulation ranges by categorizing the gene expression data generated from high-throughput microarray or RNA sequencing. Housekeeping genes are generally used to categorize gene expression data. 

Housekeeping genes are essential for the existence of the cell, regardless of their specific role in the tissue or organism. Housekeeping genes are expressed in all cells of an organism regardless of conditions (normal or pathophysiological), tissue type, developmental stage, cell cycle status, or external signals. Unlike in qRT-PCR, housekeeping genes are not generally used for normalization in RNA sequencing analysis. Therefore, the variation in gene expression measurements due to different sample preparation techniques is not accounted for in the RNA expression analysis. A set of stably expressed housekeeping genes in particular tissue types should be used for the corresponding research. For instance, a set of housekeeping genes were used for NSCLC [56,57,58,59,60], including *ACTB*, *B2M*, *CDKN1B*, *ESD*, *FLOT2*, *GAPDH*, *GRB2*, *GUSB*, *HMBS*, *HPRT1*, *HSP90AB1*, *IPO8*, *LDHA*, *NONO*, *PGK1*, *POLR2A*, *PPIA*, *PPIH*, *PPP1CA*, *RHOA*, *RPL13A*, *SDCBP*, *TBP*, *TFRC*, *UBC*, *YAP1*, and *YWHAZ* to define the threshold of gene expression level in multi-omics regulatory network studies [48,61]. Specifically, the total percentage of up-regulated and down-regulated samples was fixed for all the housekeeping genes to be 30%, and the average standard deviation of the normal range for the selected housekeeping genes was calculated. This average standard deviation was applied to all other genes in the genome to define their normal, up-regulation, or down-regulation ranges [48,61]. “Half SAM score” is recommended for differential gene expression analysis of data generated from microarrays and next-generation sequencing (NGS) [62]. DEseq2 is commonly used for fold change and differential gene expression analysis of NGS data [63]. 

#### 2.1.3. Categorization of Protein Regulation

Protein expression represents how proteins are synthesized, modified, and regulated in an organism. The synthesis and regulation of proteins depend on the functional requirements in the cell. The blueprint for proteins is stored in DNA and decoded by a highly regulated transcriptional process that produces messenger RNA (mRNA). The information encoded by mRNA is subsequently translated into proteins as functional units of biological processes. Protein expression data generated from AQUA [56] and Nano-LC-MS/MS [64] are often log-transformed for differential expression analysis and Cox survival modeling. 

The up-regulation, normal, and down-regulation ranges of protein expression also need to be defined, similar to gene expression. In a regulatory network analysis of NSCLC tumors [64], the categorization of protein regulation was performed by using the normal range defined with NSCLC housekeeping genes [56,57,58,59,60], including *B2M*, *ESD*, *FLOT2*, *GAPDH*, *GRB2*, *HPRT1*, *HSP90AB1*, *LDHA*, *NONO*, *POLR2A*, *PPP1CA*, *RHOA*, *SDCBP*, and *TFRC*, based on their protein expression in NSCLC tumors and non-cancerous adjacent tissues in Xu’s cohort [65]. The total percentage of up-regulated and down-regulated samples was fixed for all the housekeeping proteins, and the average standard deviation of the normal range for the selected housekeeping proteins was calculated and applied to all other proteins in the genome to define their normal, up-regulation, or down-regulation ranges [64].

### 2.2. Single-Cell Muti-Omics Data Processing

Each cell type has its distinct function. The single-cell analysis allows the study within a cell population to reveal how cell networks function [66,67]. Ginkgo [68] is an open-source web-based platform for single-cell CNV analysis. Single-cell transcriptomics simultaneously measures the gene expression level of individual cells in a given population [69]. Single-cell whole-genome analyses by Linear Amplification via Transposon Insertion (LIANTI) can generate sufficient DNA for next-generation sequencing [70]. In processing the single-cell gene expression data from Illumina HiSeq 2000, gene features are counted with the *featureCounts* method [71] based on the Gencode v19 transcriptome annotation. In processing the data from Illumina HiSeq 4000, the reads are mapped with *STAR* aligner [72] based on human genome reference GRCh38, and SAMtools [73] is used to sort and index the mapped reads.

The dropout phenomenon, i.e., the RNA in the cell is not detected due to limitations of current experimental protocols, is severe in single-cell transcriptomic data. As a result, a large number of genes are expressed with a value of 0 in many cells. This makes it difficult to classify single-cell gene expression as in bulk tumors, and the housekeeping gene technique described above cannot achieve usable results. Thus, single-cell gene expression data is generally classified into two categories, “not expressed” for genes with a feature count of 0, and “expressed” for genes with a future count greater than 0 in regulatory networks [74]. DEsingle [75] in Bioconductor is a common method for single-cell differential expression analysis. 

## 3. Common Systems Biology Software and Data Resources

### 3.1. Pathway Analysis

Molecular pathway analysis is important to translate multi-omics analysis to drug discovery [76]. Once a list of genes is identified from a study, gene set enrichment analysis can be performed to examine the relevant biological processes and canonical pathways. Enrichment is the process of classifying genes according to a priori knowledge. The following tools are used for pathway analysis.

GSEA is an online tool to evaluate the over-representation of a gene list in a comprehensive database MSigDB [77]. The input to GSEA is a gene expression matrix in which the samples are divided into two sets. All genes are first sorted from largest to smallest based on the processed differential expressions, which are used to represent the trend of gene expression changes between the two sets. GSEA analyzes whether all genes in a gene set are enriched at the top or bottom of a ranked list for a biological process. If they are enriched at the top, the gene set is considered overall up-regulated in this biological process. Conversely, if they are enriched at the bottom, this gene set is considered overall down-regulated in this biological process.

ToppFun in ToppGene Suite is a one-stop portal for enrichment analysis and candidate gene prioritization based on functional annotations and protein interaction networks [78]. ToppFun provides enrichment analysis of pathways, gene families, cytobands, drugs, diseases, etc. The input to ToppFun is a list of genes. The outputs include significant functional enrichment results with information such as *p*-values, FDRs, etc.

Qiagen Ingenuity Pathways Analysis (IPA) is an online pathway analysis tool incorporating curated molecular interactions and their involvement in diseases with confirmed information retrieved from scholarly publications. Using these data, it is possible to map interactions among a list of genes in various pathological conditions, such as cancer and immunological diseases.

Adviata iPathwayGuide computes the over-representation of an input gene list in a pathway or disease using Fisher’s method. Multiple hypothesis testing is applied using FDR or Bonferroni corrections. The enrichment analysis utilizes pathways and diseases from the Kyoto Encyclopedia of Genes and Genomes (KEGG) database [79,80], gene ontologies (GO) from the Gene Ontology Consortium database [81], and miRNA-mRNA target pairs from the miRBase and MICROCOSM databases [82]. Experimentally confirmed microRNA targets can be retrieved from TarBase [83]. 

### 3.2. Proliferation Assays

Cancer cells have high rates of cell division and growth, and are very prolific. The DepMap portal provides genome-scale CRISPR-Cas9/RNA interference (RNAi) screening data in Cancer Cell Line Encylopedia (CCLE). The dependency scores in CRISPR-Cas9 [84] knockout and RNAi [85] knockdown screening data measure a gene’s impact on proliferation. Essential genes significantly impact the cellular growth in a cell line in knockout/knockdown assays; otherwise, they are defined as nonessential. Gene knockout/knockdown effects, represented with dependency scores, are normalized such that the median dependency score of the non-essential genes is 0, and the median dependency score of the essential genes is –1 in each cell line. Negative dependency scores indicate the cancer cell line growth is highly dependent on the gene; positive dependency scores indicate the cell line grows faster after the gene is knocked out or knocked down. A normalized dependency score less than –0.5 is considered a significant effect in CRISPR-Cas9 knockout or RNAi knockdown.

The current single-cell technologies, including single-cell sequencing and CRISPR-Cas9/RNAi screening, have not been widely adopted. Recent studies explored editing immune cells using CRISPR-Cas9 [86,87,88,89]. Nevertheless, there is a lack of single-cell genome-scale CRISPR-Cas9/RNAi screening data for broad research and clinical applications.

### 3.3. Stromal and Immune Infiltration and Cell Activity

The extracellular matrix, soluble chemicals, and tumor stromal cells constitute the tumor microenvironment. The formation of the tumor microenvironment will result in the chemotaxis of numerous immune cells (e.g., macrophages, T cells, etc.) that form part of the tumor microenvironment. In the tumor microenvironment, immune cells and stromal cells are the two main non-tumor components, which are of great potential for cancer diagnosis and prognosis assessment.

The Estimation of STromal and Immune cells in MAlignant Tumours (ESTIMATE) [90] predicts tumor purity and infers the stromal and immune infiltration in tumor tissues. The function *estimateScore* of the *ESTIMATE* package in R computes the stromal score and immune score in each sample using transcriptomic data. 

The xCell tool [91] predicts the levels of 64 immune and stroma cell types based on gene expression data. The xCell scores for patient samples can be calculated using single-sample gene set enrichment analysis (ssGSEA) to analyze the immune microenvironment. Low xCell scores indicate the cell type has similar levels across all samples; whereas high xCell scores indicate the cell type has different levels across all samples.

TIMER 2.0 [92,93,94] and CIBERSORTx [95] are comprehensive resources for systematically analyzing the immune infiltration in tumors. They provide the abundance of immune infiltration estimated by a variety of immune deconvolution methods. TIMER 2.0 [92,93,94] and CIBRSORTx [95] compute the association of gene expression and immune infiltration in multiple cell types including myeloid dendritic cells, macrophages, neutrophils, CD4+ T cells, CD8+ T cells, B cells, etc. using a variety of immune deconvolution methods. Microenvironment Cell Populations-counter (MCP-counter) [96] quantifies the absolute abundance of eight immune and two stromal cell populations in heterogeneous tissues using transcriptomic data. MCP-counter estimates immune infiltrates across healthy tissues and non-hematopoietic tumors in human samples.

### 3.4. Drug Discovery and Repurposing

LINCS L1000 Connectivity Map (CMap) [26,27] provides an online tool to identify functional pathways and drugs based on gene expression signatures of up-regulated or down-regulated genes. CMap incorporates over 1.5M transcriptomic profiles from the treatment of ~5000 small molecules and ~3000 genetic reagents in multiple cell types. A hypothesis is considered valid for further investigation with a *p*-value < 0.05 and a connectivity score > 0.9. The selected compounds can be further analyzed with the drug screening data to discover potential repositioning drugs [48,61,74].

Drug screening data from PRISM [97] and GDSC1/2 [98,99,100] datasets contain drug activity data in CCLE. Multiple doses were tested for each drug. Cell lines are considered resistant to a drug if the IC_50_ or EC_50_ values are higher than the maximum dose; cell lines are considered sensitive to a drug if the IC_50_ or EC_50_ values are lower than the minimum dose. Using the mean ± 0.5 standard deviations of the drug sensitivity measurements, the remaining cell lines can be categorized into three groups, including sensitive, partial response, or resistant [101,102]. This in vitro drug sensitivity categorization is corresponding to RECIST 1.1 (i.e., complete response, partial response, and stable disease/disease progression) in evaluating therapeutic responses in patient solid tumors [103]. 

## 4. Common Approaches to Molecular Network Inference

Molecular networks have been widely used to understand multicellular functions in disease [104] and elucidate drug response from modulators to targets [105]. Artificial intelligence/machine learning methods are needed to construct multi-omics genome-scale networks. This section reviewed common approaches for network inference in terms of computational efficiency, scalability, and accuracy. 

### 4.1. Relevance Networks

Relevance Networks mainly construct gene regulatory network (GRN) models by calculating the associations between genes. This method considers that genes with similar expression profiles may interact with each other and therefore may have similar functions [106]. If the expression value of gene A is increased and the expression value of gene B is simultaneously increased or decreased, the relationship between the two genes can be detected and modeled. The regulatory relationship can also be inferred by the transcriptional dependence between them. The main idea of the correlation detection method is that for a predetermined threshold if the association between genes is higher than the threshold, the genes will be connected by edges in the network. Two genes are more related if they have the same or similar regulatory mechanisms, especially for target genes of the same transcription factor or genes on the same biological pathway. The relevance between genes can be inferred with the following metrics.

#### 4.1.1. Pearson Correlation Coefficient (PCC)

PCC [33] is a linear correlation coefficient, which reflects the degree of linear correlation between two variables. Let X and Y be two random variables, $PCC\left(X,\;Y\right)$ is defined as:(1)$$PCC\left(X,\;Y\right)=\frac{\sum_{i}\left(X_{i}-\bar{X_{i}}\right)\left(Y_{i}-\bar{Y_{i}}\right)}{\sqrt{\sum_{i}{\left(X_{i}-\bar{X_{i}}\right)}^{2}}\times \sqrt{\sum_{i}{\left(Y_{i}-\bar{Y_{i}}\right)}^{2}}}$$
where $\bar{X_{i}}$, $\bar{Y_{i}}$ are the mean values of *X* and *Y*, respectively. $PCC\left(X,\;Y\right)\;$ takes values between −1 and 1. When $PCC\left(X,\;Y\right)$ is −1 or 1, it means that the two variables are completely correlated; when $PCC\left(X,\;Y\right)$ is 0, the two variables are linearly uncorrelated. Figure 1 shows a simple example of constructing a network model using PCC.

Weighted gene correlation network analysis (WGCNA) is a typical method for constructing gene co-expression regulatory networks with PCC [107], where genes are first divided into clusters using hierarchical clustering, and highly co-expressed genes in each cluster are connected by correlation values. Genomic networks are established after the interrelationships of every pair of genes have been determined. Various correlation networks have been implemented for multi-omics analysis. MiBiOmics [108] implements WGCNA in R as a Shiny app for multi-omics network analysis and visualization. OmicsAnalyst [109] system models correlation networks and is hosted on Google Cloud. CorDiffViz [110] is an R package to construct and visualize multi-omics differential correlation networks. In addition to Pearson’s correlation, CorDiffViz utilizes rank-based correlation metrics coping with non-Gaussian observations commonly present in omics data for more robust inferences of differential correlations. The outputs are automatically saved to a local directory by calling a single R function *viz()* with some specified parameters. The users can then visualize the results by opening the HTML file in a browser. 

Since PCC only needs to calculate the similarity of expression profiles between genes, it has the advantage of low computational time and space complexity. It is therefore able to cope with large-scale data and can be applied to both continuous and discrete data, but not categorical data. Since PCC can only measure the linear relationship between nodes, it is only capable of analyzing genes with similar expression profiles. Moreover, PCC is vulnerable to noise and random perturbations, which makes it inaccurate and less robust.

#### 4.1.2. Gaussian Graphical Models (GGM) 

GGM is an undirected probabilistic graphical model that assumes gene expression data follow a multidimensional normal distribution. A partial correlation coefficient matrix between genes is first calculated, and then the edges of the network are selected by testing whether each element of the partial correlation coefficient matrix is significantly different from zero.

For a network containing *n* genes with expression levels denoted as $x_{1},\;x_{2},\;\ldots ,\;x_{n}$, assuming the genes are joint normally distributed, the partial correlation coefficient is:(2)$$\rho_{ij}=Corr\left(x_{i},\;x_{j}|x_{-\left(i,\;j\right)}\right)$$
where $x_{-\left(i,\;j\right)}=\left\{x_{k}|1\leq k\neq i,\;j\leq n\right\}$. $\rho_{ij}\;\neq 0$ means the two genes are conditionally dependent so that there is an edge between them. The partial correlation coefficient can be expressed as the inverse of the covariance matrix as follows:(3)$$\rho_{ij}=-\frac{\sigma_{ij}}{\sqrt{\sigma_{ii}\sigma_{jj}}}$$
where $\sigma_{ij}$ is the element of the inverse covariance matrix. However, if the number of genes *n* is large but the sample size is small, the covariance matrix cannot be obtained. To avoid the calculation of the covariance matrix, some methods were proposed based on low-order partial correlation analysis [111,112,113].

The advantage of GGM is that it can eliminate a lot of indirect connections between genes to facilitate further analysis. The disadvantages are (1) the edges of the network it constructs are undirected and cannot infer causality, and (2) the static model cannot reflect the dynamic behaviors in GRNs.

#### 4.1.3. Mutual Information (MI) 

To improve the limitations of correlation coefficients in association-based methods, information theory-based methods have been proposed for the construction of GRNs. Mutual information (MI) [114] is usually used to describe the statistical correlation between two systems or to reflect the amount of information embedded in one system about the other system using entropy [115,116]. According to the definition of entropy in information theory, the mutual regulatory information between genes can be analyzed from an information theory point of view, and the gene expression information can be quantified by using the Shannon evaluation of information entropy [117]. The entropy of a gene expression pattern is a measure of the information contained, and the model describes the association of genes in terms of entropy and mutual information.

The entropy of a gene expression pattern X is a measure of the amount of information it contains and is calculated as:(4)$$H\left(X\right)=-\underset{x}{\sum }p\left(x\right)\log_{2}p\left(x\right)$$
where $p\left(x\right)$ is the probability of X=x. The larger the entropy value, the more the gene expression level tends to be randomly distributed. Let $p\left(x,\;y\right)$ be the joint probability when X=x and *Y*
=y, then the joint entropy of X and Y is defined as: (5)$$H\left(X,\;Y\right)=-\underset{x}{\sum }\underset{y}{\sum }p\left(x,\;y\right)\log_{2}p\left(x.\;y\right)$$

From this, we can obtain the MI of the random variables *X* and *Y* as:(6)$$MI\left(X,\;Y\right)=H\left(X\right)+H\left(Y\right)-H\left(X,Y\right)$$

A high MI indicates a close relationship between two genes and a low MI indicates their independence [118]. To construct a gene association network, the MI is used to (1) calculate the degree of association between all gene pairs, (2) define the existence of associations between gene pairs that pass the pre-set threshold, and (3) to connect these gene pairs with edges [119].

MI is a widely recognized metric for quantifying statistical association [120]. The most important advantage of MI is its ability to infer nonlinear relationships between genes accurately and efficiently [121,122,123]. Secondly, MI can handle large-scale data with a limited sample size [124,125,126]. The disadvantage is that MI may overestimate the interaction relationships between genes and the constructed networks tend to contain many false-positive edges. To reduce the false positive edges, the influence of other genes can be analyzed and eliminated when calculating the association degree of two genes, i.e., Conditional Mutual Information (CMI). CMI was introduced to delete these false positive edges [118,127]. However, CMI appears to underestimate the regulatory relationships between genes in some cases, increasing false negative network edges. To address these problems, Zhang et al. [128] proposed the CMI2NI algorithm, which reduces this error by introducing the concept of relative entropy by calculating the Kullback-Leribler divergence.

The association network model can only obtain whether two genes are associated or not, but cannot infer the specific regulatory relationship. To distinguish direct and indirect effects, various optimization methods based on information theory have been proposed. The Algorithm for the Reconstruction of Accurate Cellular Networks (ARCANE) by Margolin et al. [121] employs an information theory-based approach to constructing association networks by using the Data Processing Inequality (DPI) constraint. If the data processing imbalance exceeds a certain threshold, ARCANE evaluates all possible gene triplets and prunes the least significant edge in each triplet with the smallest MI among the corresponding genes. ARACNE is a relatively conservative network construction method that retains the majority of edges inferred from the network. The Context Likelihood of Relatedness (CLR) algorithm [111] proposes an adaptive background correction step to remove erroneous correlations. CLR estimates paired MI values for all gene pairs and then converts the MI values into *z*-scores for comparison with the sample distribution to estimate the statistical possibility of the specific gene pair. Maximum relevance/minimum redundancy (MRMR) used by MRNET [129] can infer gene interactions. The MRMR algorithm is used to choose the ideal subset of regulators, which initially treats one gene as the target gene and the rest genes as its potential regulators. C3NET [130] retains only the core causality of the network, i.e., only the MI of the gene pair that is higher than the MI with any other gene in the genome for both genes, and then the connection between these pair of genes will be established.

MI constructs undirected networks, and most applications require known gene regulation to assume the directionality between genes. Therefore, the wide use of such methods is limited by the available a priori information of the data.

### 4.2. Bayesian Belief Networks (BBNs)

The Bayesian belief network (BBN) model is a probabilistic graphical model describing the conditional structural independence between random variables and is used to construct networks by establishing a joint probability distribution of nodes using the Bayesian theory. The concept of the probabilistic graphical model was first proposed and applied to intelligent systems [131,132,133]. Hartemink et al. [134] proposed Bayesian network-based GRN models around 2000. The Bayesian theory was combined with graph theoretic models to quantitatively model the general properties of gene regulation [135].

Learning the BBN structure from data is all about finding a network that fits best to a given dataset. Suppose $B_{1},\;B_{2},\;\ldots ,\;B_{n}$ denote random events in the sample space and $P\left(B_{i}\right)$ can be estimated based on previous data analysis or prior knowledge, then $P\left(B_{i}\right)$ is said to be the prior knowledge; the probability of $B_{i}$ occurring in the case of event A is the posterior, i.e., $P\left(B_{i}|A\right)$. As the sample information keeps changing, the posterior probabilities are constantly updated. The previous posterior probability will be used as the prior probability when obtaining the new posterior probability. This is a process of constant updating and iterative adjustment. 

A BBN consists of two parts: the network structure and the conditional probability distribution, which can be defined as a binary group: $B=\left(G,\;P\right)$. $G=\left(N,\;E\right)$ is a Directed Acyclic Graph (DAG) [136]. N is the set of nodes, and each node $X_{i}$ can be regarded as a variable taking discrete or continuous values. E is the set of directed edges, and each edge represents a directed probabilistic dependency between two nodes. The degree of dependency is determined by the conditional probability. P is a set of conditional probability distributions: $P=\left\{P\left(X_{i}|parent\left(X_{i}\right)\right)\right.:\;X_{i}\in \left.N\right\}$ [137], where $parent\left(X_{i}\right)$ is the parent nodes of $X_{i}$ in graph G. The edge pointing from $X_{i}$ to $X_{j}$ indicates that $X_{i}$ is the parent node of $X_{j}$. The Markov assumption is implicit in BBNs that the probability of each node is related to its parent node only, i.e., each node is independent of its non-child nodes when the parent nodes are known [138]. Based on this conditional independence property, by applying the chain rule of probability and conditional independence, the joint probability distribution of the specified settings in the Bayesian network G can be expressed in the form of a product:(7)$$P\left(X_{I},\;X_{2},\;\ldots ,\;X_{n}\right)=\overset{n}{\underset{i=1}{\prod }}P\left(X_{i}|parent\left(X_{i}\right)\right)$$

The BBN model can describe the state of a gene in both discrete and continuous data, which provides an intuitive and simple way to understand and present GRNs. Figure 2 depicts an example of a discrete BBN. According to Equation (7), the probability that the state of all three (A, B, and C) nodes are 1 is: $P\left(A=1,\;B=1,\;C=1\right)=P\left(B=1\right)\times P\left(A=1|B=1\right)\times P\left(C=1|A=1,\;B=1\right)=0.6\times 0.7\times 0.7=0.294$.

The usage of BBNs requires the computation of conditional probabilities between child nodes and all their possible parent nodes, which grows exponentially in computational time as more variables are incorporated. It has been shown that finding the optimal graph for BBNs is an NP-hard problem [139], which poses a tremendous challenge to constructing complete gene regulatory networks for higher organisms such as humans [140]. To solve this problem, Campos et al. [141] proposed a method based on structural constraints that can reduce the search space by inferring the maximum number of potential parents of a node. Liu et al. [142] designed the Local Bayesian Networks model by (1) first constructing the initial graph with mutual information and conditional mutual information, (2) then splitting the initial network by the K-nearest neighbors algorithm to reduce the search space, (3) using Bayesian networks to build small the sub-networks, and (4) finally integrating the generated sub-networks. 

BBNs can process random data, fuse different types of data and prior information to introduce a suitable network structure, and can handle incomplete noisy data and hidden variable data [143]. The BBN model is highly interpretable and the results are accurate, but it is computationally complex. Hence, it is less capable of handling large-scale data and needs to develop appropriate methods to reduce the search space. When applied to GRNs, BBNs consider the noise of the gene expression data itself as well as the stochastic nature and allow the use of Bayesian theory to incorporate some a priori biological knowledge, such as heterogeneous information [137], in deciding gene relationships. 

CBNplot [144] is an R package that uses biological pathway information curated by enrichment analysis to construct and visualize BBNs. The structural inference in CBNplot is based on the bootstrap method of the R package *bnlearn*, which uses preprocessed gene expression data to infer BBNs, and uses eigengene as the expression value for pathway inferences. The results of the CBNplot highlight the interactions between genes and pathways through knowledge mining and visualization. CBNplot can be installed in R through *devtools*. The algorithms in the R package *bnlearn* are implemented with C++ in *BayesNetty* [145,146]. Networks are drawn with the *igraph* R package. TETRAD IV [147] is another implementation of BBNs. 

There are major limitations of BBNs. First, BBNs identify regulatory networks as directed acyclic graphs (DAG) that do not include feedback loops. Second, BBNs do not take into account the dynamics of regulatory relationships [133], although feedback loops and dynamics are very important features of regulatory networks.

### 4.3. Dynamic Bayesian Networks (DBNs)

To capture the dynamic characteristics in GRNs and the information on loop interactions between genes, dynamic Bayesian networks (DBN) were proposed to consider the time-delayed nature of gene regulation and incorporate the dimension of time information in BBNs. The value of a random variable in DBN is determined by the previous time point, and DBN is the transformation process of the random variables at all possible random discrete points [148,149]. The DBN structure is modeled at discrete time points *t*. Similar to the assumption of the BBN, if $X_{t}$ is the expression of *n* genes at time points *t*, the DBN can be described as:(8)$$P\left(X_{t}|X_{t-1}\right)=\overset{n}{\underset{i=1}{\prod }}P\left(X_{i,\;t}|parent\left(X_{i,\;t}\right)\right)$$
where $X_{i,\;t}$ is the expression value of the gene $X_{i,}$ on time slice *t*, and $parent\left(X_{i,\;t}\right)$ is the set of its parent nodes. Figure 3A represents a static BBN and Figure 3B represents a DBN. In the static BBN (Figure 3A), the loop A→B→C→A is not allowed, but this feedback mechanism can be represented in the DBN (Figure 3B). 

Smith et al. [150] used the DBN model to analyze microarray data, combining the negative feedback of gene regulation with the time delay factor, so it is necessary to use different nodes in the network to represent the expression of the same gene. Song et al. [151] proposed a new data integration model on DBN by combining a priori knowledge of the relationship between microarray data and genes to construct GRNs using parallel algorithms.

### 4.4. Ordinary Differential Equation (ODE) Based Networks

Differential equation models use continuous variables to describe changes in gene expression values as a function of other genes and environmental influences, and it captures the dynamics of GRNs in a quantitative form. The flexibility of differential equation modeling enables the representation of more complex relationships between components. Ordinary Differential Equations (ODEs) are often used to model GRNs. Differential equation models regard the expression level of a gene as a function of time and therefore require the use of time-series data when constructing a GRN. In the process of using ODE to construct GRNs, each differential equation represents the relationship between target genes and their regulatory factors, and the corresponding parameters determine the topology of the network and the interrelationships between genes. The ODE of GRN is expressed as:(9)$$\frac{dx_{i}}{dt}=f_{i}\left(x\right),\;1\leq i\leq n$$
where $x_{i}$ represents the expression level of gene $X_{i}.$ $X_{1}$, …., $X_{n}$ are the *n* genes that affect gene $X_{i}$, and $x={\left[x_{1},\;\ldots ,\;x_{n}\right]}^{T}$ is their expression levels. $\frac{dx_{i}}{dt}$ represents the rate of change of the expression level of gene $X_{i}$ at moment t in the GRN modeling. $f_{i}\left(x\right)$ illustrates the mode of action and the regulatory mechanism between genes, i.e., the structure of the regulatory network. The function $f_{i}\left(x\right)$ can be linear, segmented linear, pseudo linear, or continuous nonlinear functions. $f_{i}\left(x\right)$ in its simplest form is a linear function and can be expressed as:(10)$$\frac{dx_{i}}{dt}=\underset{j}{\sum }\omega_{ij}x_{j}+b_{i},\;1\leq i\leq n$$

The relationship between the genes in the regulatory network can be expressed by the parameter $\omega_{ij}$, for which the activation, repression, and no-regulation relationships take values of positive, negative, and 0, respectively. $b_{i}$ denotes the basal activity of the gene $X_{i}$. Linear differential problems can be solved using singular value decomposition, least squares regression, or likelihood-based approaches [152,153]. 

However, the regulatory relationships in cells are not simply linear [154] and can be inscribed using nonlinear regulatory functions $f_{i}\left(x\right)$. The disadvantage of nonlinear functions is the computational difficulty and the high computational cost of finding the solution to the differential equation. Moreover, the number of samples is usually too small compared to the number of genes, resulting in a non-singular matrix that will have multiple solutions satisfying the differential equation, which in turn requires the selection of reasonable model parameters from multiple solutions. Therefore, the search space of the nonlinear model structure needs to be strictly limited. Sakamoto et al. [155] used genetic programming to identify small-scale networks by fitting a polynomial function f; Spieth et al. [156] used different search mechanisms such as evolutionary algorithms to infer small networks. 

The advantages of ODE modeling are: (1) it is powerful and flexible; (2) it facilitates the description of complex relationships in GRNs; and (3) it is especially suitable for genes with periodic expression. ODE models are mathematically well expressed and have great potential in the analysis of local GRNs. In addition, ODEs can be used to study the effects on gene expression levels by changing environmental variables, introducing new variables, etc., and comparing the changes in the weight matrix before and afterward.

The disadvantages of differential equations are: (1) the parameters in the model are difficult to estimate, and (2) it is hard to obtain a globally optimal solution. In large networks, the ODE model is limited by sample size requirements, lacks robustness to noisy data, and does not capture the stochastic information contained in gene expression data very well. In the absence of constraints, the ODE system will have an infinite number of solutions. Therefore, to determine the appropriate ODE model and to perform accurate parameter estimation, a thorough study of the nature of the f function and the definition of reasonable constraints based on prior knowledge are required.

### 4.5. Boolean Networks

The Boolean network model, first introduced by Kauffman in 1969 [157], is a dynamic discrete model in which the network nodes have synchronous relationships [158]. The Boolean network model is one of the simplest models to reveal GRNs, which treats genes as logical elements [159]. Individual genes can be represented by Boolean variables regardless of whether they are expressed or not. The Boolean network model abstracts the expression level of a gene by clustering or threshold discretization into two states: on and off, the state “on” indicating that the gene is expressed (or overexpressed state) and the state “off” indicating that the gene is not expressed (or low expression state). The interactions in Boolean networks between genes must follow Boolean rules. A Boolean network contains *n* nodes (representing genes in GRNs) in the repressed or expressed states (i.e., 0 or 1), which are connected by the logical operators “*and*”, “or”, and “not” [160]. The expression level of a given gene is obtained by a Boolean function on the expression levels of multiple genes associated with that gene, and the states of all genes are updated using a synchronous update mechanism. The challenge of Boolean network construction for GRN lies in finding the appropriate Boolean function for each gene so that the model can accurately interpret the observed data.

A Boolean network is a directed graph, denoted by *G(N, E)*, *E* is the set of directed edges, where each node *Xi* ∈ *N* is determined by a function. The next state at *t* + 1 of the network can be represented by all inputs and the functions of the nodes at a time point *t*:(11)$$X_{i}\left(t+1\right)=B_{i}\left[X_{1}\left(t\right),\;\ldots ,\;X_{n}\left(t\right)\right]$$

$X_{i}\left(t\right)$ represents the expression level of gene i at moment t. The function $B_{i}$ represents the Boolean function of the whole network for gene i. The interaction relationship between genes is represented by Boolean functions, and Boolean rules are expressed in the form of truth tables. Figure 4A depicts a simple Boolean network $G\left(N,\;E\right)$, and Figure 4B represents the state transition corresponding to the linkage graph $G^{\prime }\left(N^{\prime },\;E^{\prime }\right)$ of Figure 4A. The truth table of this network is shown in Figure 4C.

Boolean networks simplify the actual GRNs, providing a framework to describe the complex interactions between genes in GRNs in a biological context. Boolean networks emphasize the underlying global network rather than a quantitative biochemical model. Boolean functions can find possible gene interaction relationships, which can be used as a basis for modeling real gene regulatory networks.

The disadvantage of Boolean networks is their imprecision. Boolean networks can only be represented as a crude qualitative model that portrays the interactions between genes by combining fixed logical rules. It is difficult to accurately describe the real GRN enumerating all possible logical operations, so the Boolean network can only be used as the basis for modeling the real GRN. The update of the network state in the Boolean model is synchronous. However, biological networks are typically asynchronous. Boolean network modeling discretizes gene expression levels into two simple values. However, in real biological systems, gene expression is not a simple state, but continuous. When discretizing gene data, it will inevitably result in the loss of many important expression information [161], which can largely affect the accuracy of the model. Moreover, gene expression regulation should have at least three states: up-regulated, normal, or down-regulated, and its discretization is a difficult process. The setting of the threshold is crucial to determine the state of the node, and errors in the threshold setting will directly lead to changes in the gene state. It in turn will lead to inaccurate inferences, which is a common drawback of discrete models. 

Liang et al. [162] first proposed to predict possible GRN structures from gene expression data using Boolean networks and developed a Boolean network-based software Reverse Engineering Algorithm (REVEAL) by considering the information entropy between nodes to help build the network structure. Kim et al. [163] proposed to utilize chi-squared tests to eliminate uncorrelated edges between nodes to accelerate the search for the optimal network structure. Due to the stochastic nature of biological systems and the noise contained in gene expression data, Boolean networks as deterministic models are not able to capture network regulatory relationships accurately. To solve this problem, a combination of the Boolean network and Markov chain was developed into the Probabilistic Boolean Network (PBN) model [164], which is a more flexible topology that adds stochasticity to the original network and can better handle the uncertainties among genes in the probabilistic framework. Boolean networks can be combined with MI to infer the structural and dynamic relationships between genes for time-series data [165]. The Single Cell Network Synthesis toolkit (SCNS) [166] is a computational tool for reconstructing and analyzing executable models from single-cell gene expression data. SCNS constructs a state transition graph of binary expression profiles using single-cell qPCR or RNA sequencing data acquired over the entire time course. An asynchronous Boolean network model is built by searching for rules that drive the transition from early to late cell states and thus reconstructing Boolean logical regulatory rules. 

### 4.6. Boolean Implication Networks

The Boolean implication is the logical relationship between two Boolean variables, where the state of one variable can imply the state of the other variable. Boolean implication networks were first proposed by Sahoo et al. in 2008 [167] for building genome-wide gene relationship networks based on microarray data. The nodes in Boolean implication networks are genes and the edges are implication relationships. The implication is an if-then rule. For example, “if gene *A* is expressed high, then gene *B* is expressed high” which can be also expressed as “*A* high implies *B* high”. The Boolean implication network automatically sets a separate threshold for each gene, which is used to classify the expression of a gene as “low” or “high”. Then, the Boolean implications will be identified between each pair of genes in the whole genome.

There are six possible Boolean relationships in the Boolean implication network, including four asymmetric relationships: “*A* high ⇒ *B* high”, “*A* high ⇒ *B* low”, “*A* low ⇒ *B* high”, and “*A* low ⇒ *B* low”; two symmetric relationships: if “*A* high ⇒ *B* high” is accompanied by “*B* high ⇒ *A* high”, then gene *A* and gene *B* are “Boolean equivalent”, if “*A* high ⇒ *B* low” and “*B* high ⇒ *A* low” at the same time, then gene *A* and gene *B* are “opposite”.

The process of establishing Boolean connections between two genes *A* and *B* is shown in Figure 5. First of all, each gene has a threshold *t* derived using the StepMiner algorithm [168], and the interval of t±0.5 is called “intermediate” (the gray areas in Figure 5). The values in the “intermediate” area may be misclassified, so the values in the “intermediate” range do not participate in the network creation process. Next, among the four quadrants consisting of high-low expressions of gene *A* and gene *B*, we need to check which one is significantly sparser than the other quadrants. The sparsity can be calculated using the statistic and error rate (Equations (12) and (13)). For example, we want to detect the sparsity in quadrant IV (i.e., *A* high *B* low) and thus infer the implication rule of *A* high ⇒ *B* high. Let $a_{I},\;a_{II},a_{III},\;a_{IV}$ be the number of values in each quadrant.
$$total=a_{I}+a_{II}+a_{III}+a_{IV}$$
$$n_{A\;high}=a_{I}+a_{IV}$$
$$n_{B\;low}=a_{III}+a_{IV}$$
$$expected=\frac{n_{A\;high}+n_{B\;low}}{total}$$
$$observed=a_{IV}$$
(12)$$statistic=\frac{expected-observed}{\sqrt{expected}}$$
(13)$$error\;rate=\frac{1}{2}\left(\frac{a_{IV}}{a_{IV}-a_{I}}+\frac{a_{IV}}{a_{IV}+a_{III}}\right)$$

If quadrant IV has a calculated statistic greater than 3.0 and an error rate less than 0.1, we will consider that rule *A* high ⇒ *B* high as significant. After the whole genome, pairwise Boolean implication rules were generated, and the Boolean Implication network was built. 

Sinha et al. [169] applied the Boolean implication on mutation, copy number, methylation, and gene expression data. Their results indicated that a large number of Boolean implications exist in the data that could not be detected by other methods. Further analysis using GSEA showed that the genes obtained through their Boolean implications also have biological significance. Their analysis showed that Boolean implications could be used for finding genes whose expression was regulated by copy number variations or DNA methylation changes. In the work of Çakır et al. [170], the combination of Boolean implication analysis with SOM metadata found relationships between genes, metagenes, and similarly behaving metagene groups, and provided a more general and functional module-oriented view of the data. 

The advantages of the Boolean implication networks are: (1) it can indicate the biological mechanisms between the pair of associated genes; (2) it is a directed graph in that each gene pair have a causal relationship with each other; and (3) it can incorporate feedback loops in the network. The disadvantage of these Boolean implication networks is all the gene variables have to be binary. Genes with an expression level in the “intermediate” range that is not up-regulated or down-regulated are removed from the analysis. Similarly, genes with normal copy numbers are also removed in the Boolean implication network modeling. This will result in considerable data loss and incomplete representation of real GRNs. 

### 4.7. Prediction Logic Boolean Implication Networks (PLBINs)

We developed Boolean implication networks [171,172] based on prediction logic [173]. Using this algorithm, Boolean implication networks are inducted with logic rules connecting two binary variables. A contingency table is created for each pair of genes (Figure 6). The cells in the contingency table show the count of samples in each situation. There are six possible Boolean implication rules, and the count of error cells for each implication rule is used to calculate the precision and the scope to select the implication rules. The implication rules and their corresponding error cells are shown in Figure 6. The scope $U_{p}$ and precision $\nabla_{p}$ for implication rule between $node_{i}$ and $node_{j}$ are calculated with the following equations:(14)$$U_{p}=U_{ij}=\frac{N_{i.}\times N_{.j}}{N^{2}}$$
(15)$$U_{p}=\underset{i}{\sum }\underset{j}{\sum }\omega_{ij}\times U_{ij}$$
(16)$$\nabla_{p}=\nabla_{ij}=1-\frac{N_{ij}}{N\times U_{p}}$$
(17)$$\nabla_{p}=\underset{i}{\sum }\underset{j}{\sum }\frac{\omega_{ij}\times U_{ij}}{U_{p}}\times \nabla_{ij}$$

Equations (14) and (16) are for multiple cells, where $\omega_{ij}=1$ for error cells, otherwise, $\omega_{ij}=0$. To select the implication rules, thresholds for precision and scope are defined by the one-tailed *z*-tests based on the sample size and a preset *z* value. If a rule has the highest scope and precision amongst all six rules, and the scope and precision are greater than the thresholds, this implication rule will be considered a significant rule. The *z* value used for network construction is at least 1.645 (95% confidence interval, *α* = 0.05, one-tailed *z*-tests).

The use of error cells enables the analysis of multivariate data in Boolean implication networks. The computational complexity of constructing genome-scale networks is *O*(*n^2^*), where *n* is the number of genes. Our PLBINs can model cyclic relations including feedback loops. PLBINs have been applied in modeling multi-omics [48,61,64] and single-cell [74] networks for the discovery of prognostic biomarkers and therapeutic targets in NSCLC. 

### 4.8. Neural Networks

As an important branch in the field of machine learning, neural networks have been applied to systems biology and bioinformatics [174,175,176]. The relationships between genes and other gene products are often so complex for simple linear models to capture. Inspired by the animal central nervous system, neural networks are an effective mathematical model to learn multilayered complex patterns in linear and nonlinear functions. These advantages allow them to capture data features well and meet the requirements of higher accuracy in modeling multi-omics GRNs.

Neural Networks consist of multiple layers of neurons that are connected with other neurons in their preceding and succeeding layers. These neurons form three basic types of layers: the input layer, hidden layer, and output layer. A basic structure of the neural network is shown in Figure 7. The neural network model passes the feature representation of each level to the next level of unit modules by combining some simple nonlinear unit modules. By combining such nonlinear modules, neural networks can automatically extract higher and more abstract features from the original data and portray a more detailed biological data structure, which can provide modeling for complex nonlinear systems [177,178]. Neural networks introduce nonlinear factors through activation functions such as the Relu function, Sigmoid function, and Tanh function.

Alipanahi et al. [179] developed the DeepBind framework to predict the sequence specificities of DNA- and RNA-binding proteins using deep learning models. The DeepBind framework consists of a convolutional neural network for feature representation learning and a fully connected prediction module for feature combination, using gradient descent and backpropagation algorithms to train the model and compute nucleic acid binding interactions from different datasets. The DeepBind framework can be applied to a wide range of datasets and can improve the predictive power compared to traditional single-domain methods. 

The Recurrent Neural Network model exhibits strong modeling capabilities with its nonlinear structure and can adaptively recognize and remember temporal and spatial patterns, which can more realistically simulate the working processes of real biological systems. Because of its ability to establish nonlinear and dynamic interactions between genes, RNN is also a well-established method for deriving GRNs with up to 30 genes [180].

Graph neural networks (GNN), as a generalization of neural networks, are deep learning architectures that can handle graph and graph-related problems, such as node classification, link prediction, and graph classification [181,182]. Graph Convolution Networks [183], a kind of GNN, migrate the traditional convolutional operations in deep learning to the processing of graph-structured data and specify them through complex spectral graph theory derivation. Its core idea is to learn the features of a node in a graph itself and the features of its neighbors and aggregate them to generate a new mapping of functions representing vectors for this node. Wang et al. [184] proposed an end-to-end gene regulatory graph neural network (GRGNN) to reconstruct GRNs in a supervised and semi-supervised framework using gene expression data. To obtain better inductive generalization, the GRN inference is formulated as a graph classification problem to distinguish whether a subgraph centered on two nodes contains a link between these two nodes. The computational complexity to construct GRGNN is exponential of the number (*h*-hop) of subgraphs [184]. Single-cell graph neural network (scGNN) was developed to provide a hypothesis-free deep learning framework for single-cell RNA-sequencing data imputation and clustering [185]. Major deep learning-based methods in cancer classification and clustering using multi-omics and single-cell data were benchmarked [186]. 

The main limitation of neural network-based GRN inference is the requirement of the training data. The network training requires benchmarks with systematically explicit, experimentally validated, gold-standard conditioning relationships. On species with complete data, the goal of inference may be easily achieved. However, it is challenging in constructing genome-scale neural network-based GRN in complex human diseases, such as cancer. Two problems exist with deep learning modeling of genomic data: (1) the insufficient amount of training data, which affects the model performance, and (2) the high data dimensionality, which leads to a huge number of model parameters and increases the training difficulty. In addition, although neural networks are very good at learning complex tasks, their internal descriptions are generally difficult to interpret, and training deeply layered models is algorithmically difficult to handle and statistically prone to over-fit. 

### 4.9. Summary of Existing Network Inference Methods

These network methods have many applications in the discovery of biomarkers [187,188,189,190,191,192,193] and therapeutic targets [194,195,196]. They are implemented in several software packages. GeNeCK [197] is a web server that allows users to build GRNs from expression data using different network construction methods, including four partial correlation-based methods: *GeneNet*, *NS*, *SPACE*, and *ESPACE*; four likelihood-based methods: *GLASSO*, *GLASSO-SF, BayesianGLASSO*, and *EGLASSO*; and two mutual information-based methods: *PCACMI* and *CMI2NI*. *EGLASSO* and *ESPACE* accept hub gene specification to improve the network results. There is also an ensemble method, *ENA*, which does not require choosing or tuning parameters so it is suitable for most users. *ENA* provides a *p*-value for each edge in the network to indicate its statistical significance. The Weka software (Version 3.8.6) [198] implements commonly used machine learning methods for classification, including radial basis function networks and Bayesian belief networks (BBNs). A summary of software tools for multi-omics processing, pathway analysis, and network inferencing is provided in Table 1. 

Despite the successful applications of these network models in classification and clustering, there are certain limitations in these methods to construct genome-scale GRNs using emerging multi-omics data. Relevance networks can only measure the linear relationship between genes and are not robust. Relevance networks cannot model categorical data such as DNA mutations or CNVs in muti-omics analysis. Bayesian networks have high computational complexity and can only be used for small and medium-sized data. The static Bayesian networks cannot represent cyclic relations such as feedback loops. The parameter learning process of ODE networks is very complex and is limited by data sample size. Other Boolean (implication) networks can only model binary variables, which do not present biological states of mutations, CNVs, or gene/protein expression. Neural networks are limited by high requirements for sample size and completeness of information in training data and the exponential complexity of the number of subgraphs, which makes it infeasible to model genome-scale multi-omics networks for complex human diseases such as cancer.

Our PLBINs overcome the limitations of other methodologies. First, PLBINs can integrate discrete CNV data and continuous gene/protein expression data seamlessly that relevance networks cannot. Second, PLBINs can model cyclic molecular interactions that the acyclic Bayesian networks cannot. Third, PLBINs have a computational complexity of *O*(*n^2^*) and can efficiently model genome-scale GRNs that Bayesian networks, neural networks, and ODE networks cannot. Finally, PLBINs can model multinary data with robust statistical tests, whereas other Boolean networks can only analyze binary variables. This is a major advantage of PLBINs because there need to be at least three biologically relevant states without losing important information in categorized CNV (amplification/normal/deletion) andgene/protein expression data (upregulation/normal/downregulation). Our PLBINs identified gene signatures that accurately predict the risk of lung cancer risk and tumor recurrence, outperforming previous studies ones in the same patient data [171,199,200,201], meanwhile, revealing more biologically relevant molecular interactions than other network methodologies in comprehensive evaluation with MSigDB [171,201]. Using our PLBINs, we developed a seven-gene signature for NSCLC prognosis and prediction of the clinical benefits of adjuvant chemotherapy in early-stage NSCLC patients, including clinical trials [56]. Our 7-gene signature is unique that it (1) works on all NSCLC histological subtypes and multiple clinical testing platforms; (2) predicts the risk of tumor recurrence; and (3) classifies NSCLC tumors from non-cancerous normal adjacent tissues [56,61,64]. 

The comparative data of our PLBINs and other methods were published previously [171,201]. The precision and false discovery rate (FDR) of the gene coexpression networks were evaluated as follows [171]. The validity of computationally derived coexpression relations was comprehensively evaluated with five gene set collections (positional, curated, motif, computational, and Gene Oncology) and canonical pathway databases in the MSigDB [77]. A coexpression relation was labeled as a true positive (TP) if both genes were present in the same gene set or pathway in any examined database. A coexpression relation was labeled a false positive (FP) if the gene pair did not share any gene set or pathway in all the examined databases. A coexpression relation was defined as non-discriminatory (ND) if at least one gene in the pair was not annotated in a database [202]. The evaluation did not include ND coexpression relations as they were not confirmatory. The precision of a gene expression network was defined as TP/(TP + FP). The precision of our identified smoking-mediated coexpression networks in NSCLC patient tumors was 100% [171]. To test the statistical significance of the network precision, the null distribution was generated in 1000 random permutations of the class labels in the test cohort. The precision of our identified smoking-mediated coexpression networks was significant at *p* < 0.001, with no TP generated in the random tests. The FDR of gene coexpression networks was defined as the average of FP/(TP + FP) in 1000 permutations. The FDR of our identified smoking-mediated coexpression networks in NSCLC patient tumors was 0.0099. In contrast, Pearson’s correlation networks did not identify any coexpression relations using the same methodology on the same datasets [171]. In the evaluation of our identified 21 NSCLC prognostic gene signatures [199] using the NCI Director’s Challenge Study [203], our PLBINs-derived gene coexpression relations from the training cohort could be successfully reproduced in both test cohorts with significantly high precision (precision = 1 for 18 gene signatures) and low FDR (FDR < 0.1) for all 21 gene signatures [201]. As a comparison, the Bayesian networks implemented in TETRAD IV [147] did not identify any coexpression relations from the training cohort that were validated in both test cohorts [201]. The Boolean implication networks by Sahoo et al. [167] did not identify coexpression with many of the major NSCLC hallmarks, making it infeasible to select marker genes with concurrent crosstalk with multiple signaling pathways as we did with our PLBINs. In the genome-scale evaluation, our PLBINs achieved significantly high precision in 1000 random tests (*p* < 0.05), whereas the precision of the Boolean implication networks by Sahoo et al. [167] was not significant in 1000 random tests (*p* = 0.21) [201]. These results demonstrate that our PLBINs are more accurate in retrieving biologically relevant gene associations, in addition to other advantages such as computational scalability and efficiency.

## 5. Hub Genes in Multi-Omics and Single-Cell Networks

Some hub genes in multi-omics networks were shown to be promising cancer biomarkers and therapeutic targets [204,205]. Nevertheless, there were insufficient genome-scale investigations on multi-omics network hub genes and their biological and clinical relevance in human cancers. Graph theory centrality metrics can characterize hub genes. Common metrics include degree centrality (in-degree and out-degree centralities) [206], eigenvector centrality [207,208,209], betweenness centrality [210,211], closeness centrality [212,213,214], and VoteRank centrality [215]. Degree centrality is simply the total number of neighbors of each node. The eigenvector centrality of a node is the sum of the centrality of its neighbors. Betweenness centrality is the frequency of a node appearing on the shortest paths of all node pairs in the entire network. Closeness centrality is the average length of the shortest paths between the node and all other nodes in the network. VoteRank centrality is selected with a voting score that is calculated by the sum of all neighbors’ voting abilities. Degree centrality and eigenvector centrality are also classified as local centrality metrics because only neighbors of each node are included in the calculation. Betweenness centrality, closeness centrality, and VoteRank centrality are categorized as global centrality metrics since the connectivity of the entire graph is used in the metrics computation. These centrality metrics are correlated in many cases [214,216]. A Python package NetworkX [217] calculates centrality metrics. 

A barrier to this systematic evaluation is the limitations of existing computational methodologies in constructing genome-scale multi-omics GRNs, as summarized above. In a recent study [218], our PLBINs were used to construct 12 genome-scale GRNs of CNV, mRNA, and protein expression in NSCLC tumors. Seven centrality metrics were correlated with NSCLC tumorigenesis measured with T-statics in differential gene/protein expression between tumors vs. non-cancerous adjacent tissues (NATs), proliferation quantified with dependency scores from CRISPR-Cas9/RNAi screening of human NSCLC cell lines, and patient survival with hazard ratios from Cox modeling of The Cancer Genome Atlas (TCGA) [218]. Hub genes in multi-omics networks involving gene/protein expression were found to be associated with oncogenic, proliferative potentials and poor patient survival. Hub genes with higher co-occurrences of CNV aberrations seemed to have tumor-suppressive and anti-proliferative properties. Regulated genes in hubs were linked to proliferative potential and worse patient survival, whereas regulatory genes in hubs were linked to anti-proliferative potential and better patient survival. Established cancer immunotherapy targets PD1, PDL1, CTLA4, and CD27 were top hub genes in most constructed multi-omics GRNs [218]. These results show that multi-omics network centrality in bulk tumors can be used in the prioritization of biomarkers and therapeutic targets. 

Similarly, our PLBINs [74] were applied to genome-wide transcriptomic profiles of B cells from tumors and NATs [219], T cells from peripheral blood lymphocytes (PBL) [220], and tumor-infiltrating T-cell gene expression data of NSCLC patients. In each cell sample, a gene was defined as expressed (with a feature count > 0) or not-expressed (with a feature count = 0). The details of single-cell network construction were provided in our previously published study [74]. The results of five single-cell co-expression networks are shown in Table 2.

We examined the centrality metrics of four established immune checkpoint inhibitors (ICIs), including *PD1, PDL1, CD27*, and *CTLA4*. Figure 8 shows the centrality distribution of the ICIs that were within the top 10th percentile in the constructed networks. *PD1* was ranked as a top hub gene in the T-cell PBL gene co-expression network in healthy donors. *CTLA4* was ranked as a top hub gene in the T-cell PBL gene co-expression network in NSCLC tumors. *CD27* was ranked as a top hub gene in the T-cell PBL gene co-expression network in NSCLC patients. These results are consistent with their functional involvements in T-cell immunity. *PDL1* was not ranked within the top 10th percentile of any of the examined centrality metrics in the constructed networks. None of these ICIs were ranked as top hub genes in B-cell gene co-expression networks in tumors or NATs.

In a previous study, we identified a gene co-expression network missing in NSCLC tumor B cells using PLBINs [74]. Genes in this network either promote proliferation in human NSCLC epithelial cells or are indicative of NSCLC patient outcomes at both mRNA and protein expression levels in bulk tumors. These network genes were associated with drug response to 10 therapeutic regimens in 135 human NSCLC cell lines. Based on this single B-cell co-expression network, we discovered tyrosine kinase inhibitor lestaurtinib as a new drug option for treating NSCLC [74]. Here, we examined if this clinically relevant single B-cell network had higher average centrality compared with 1000 random networks with the same number of genes selected from the genome. The results showed that the previously published B-cell network had significantly higher average centrality (*p* < 0.05) than 1000 random networks selected from genome-scale single B-cell networks in tumors and NATs, single T-cell PBL networks in NSCLC patients and healthy donors, and T-cell network in NSCLC tumors (Figure 9). These results support the relevance of single-cell network hub genes in tumor biology.

Our PLBIN algorithm was written in C and R and was run on Spruce Knob High-Performance Computing (HPC) Clusters. It took about 67 min for the algorithm to finish on an HPC node with 4 × 8 Intel(R) Xeon(R) CPU E5-4620 0 @ 2.20GHz 64GB memory 1TB HDD, in the case of whole-genome network construction which yielded approximately 20 million rules. The program for network centrality metrics calculation was written in Python. The running time on the HPC cluster node for a multi-omics network with 20 million edges and 12 thousand nodes for each centrality metric was provided in Table 3. The complexity of each method and the actual running time were consistent as shown in Table 3. 

A summary of concordant significant correlations between multi-omics network centrality and tumorigenesis, proliferation, and patient survival cross NSCLC cohorts reported in our previous study [218] is provided in Table 3. According to the running time (Table 2) and the concordant significant correlations (Table 4), eigenvector centrality, closeness centrality, and degree centrality were the top three performing methods in terms of computational efficiency and recapitulation of network biological and clinical relevance. Degree centrality is the best choice if genes are regulated or regulators need to be studied with in-degree or out-degree centralities, respectively. 

## 6. Integrating Multi-Omics Data with Patient Electronic Medical Records

The successful application of biomarkers and drugs requires rigorous testing in the patient population considering diverse clinical, pathological, comorbid, and demographic factors. In certain cancer types such as lung cancer, lifestyle factors including smoking, and environmental and occupational exposures, also need to be considered. Nevertheless, it is not currently feasible to conduct multi-omics and single-cell profiling in tens of thousands of patients using a well-controlled clinical study design, due to the required costs, time, and infrastructure. When a new treatment is added to the NCCN guidelines, it may take years to collect sufficient data to establish predictive biomarkers. In current biomarker studies, candidate genes are first identified from clinical cohorts of a limited number of patient samples and are then validated by leveraging public data such as TCGA. The following gaps exist in clinical applications of biomarkers: (1) Most published patient cohorts, including TCGA, do not have complete treatment information and the number of patients in specific treatment categories is very small, making it infeasible to establish predictive biomarkers of therapeutic response for clinical utility; (2) Some sequencing facilities do not have patient treatment or outcome information on all the samples they have sequenced for predictive biomarker R&D; and (3) Large-scale patient EMRs of hospital information systems or cancer registries have enough patients with comprehensive clinical information but do not have sufficient matched patient genome-scale profiles for biomarker discovery. To determine the applicability of multi-omics biomarkers in general patient populations, large-scale EMRs and genomic/transcriptomic profiles from specific patient cohorts must be combined.

By merging SEER-Medicare data, we created a unique technique to find prognostic and chemopredictive biomarkers with the potential to be used in large patient populations to fill this gap [221]. The SEER database is a compilation of registration information from specific geographic areas, which account for around 26% of the U.S. population [222]. Without additional natural language processing, the linked SEER-Medicare data are adequately annotated and prepared for computational analysis. A previous study identified chemopredictive genes by correlating mRNA expression profiles in solid tumors in the advanced cancer stage of a Serial Analysis of Gene Expression (SAGE) database with patient survival in SEER data [223]. In our previous study, a novel tumor progression indicator, combining AJCC cancer staging [224] T, N, and M factors with tumor grade was used to correlate miRNA expression in a lung squamous cell carcinoma (LUSC) patient cohort with SEER-medicare LUSC patient outcomes receiving different treatments. The identified chemopredictive miRNAs were then validated with extensive pubic data and our collected patient cohorts. Our study revealed miRNA-mediated transcriptional networks in NSCLC proliferation and progression using CRISPR-Cas9/RNAi screening data [221]. Our findings show that, in the absence of novel cohorts with tens of thousands of patients who have matched clinical outcomes and genome-scale transcriptomic profiles, extrapolation of miRNA expression from smaller cohorts to larger population-based data can serve as an additional confirmatory tool based on similarities in tumor progression. This method, in conjunction with stringent external validation, can discover prognostic and predictive biomarkers with concordant expression patterns in tumor development in sizable patient populations. 

## 7. Recommendations

Multi-omics network analysis of bulk tumors and single cells can help understand molecular mechanisms in multi-dimensional tumor immune microenvironments for the identification of clinically relevant biomarkers and effective therapeutic targets. The increasing amount of data generated with various high-throughput platforms can accelerate scientific discovery, and meanwhile, pose a challenge in harmonization and computation. To integrate genomic data such as CNV and SV generated from different sources, the genome assembly version in each dataset should be converted to hg38. To define the regulation of gene and protein expression, a set of housekeeping genes with stable expression in the studied tissue type should be used for data normalization. To construct genome-scale multi-omics regulatory networks, our Prediction Logic Boolean Implication Networks (PLBINs) have advantages over other methods in terms of computational efficiency, scalability, and accuracy [48,61,64,74]. Our recent study shows that graph theory network centralities can be used for the prioritization of biomarkers and therapeutic targets [218]. Eigenvector centrality, degree centrality, and closeness centrality are top-ranked metrics regarding time complexity and performance. Finally, multi-omic biomarkers should be integrated with patient clinical, pathological, demographic, and comorbid factors for optimal treatment selection. Our approach to integrating multi-omic profiles with large-scale patient EMRs such as the SEER-Medicare cancer registry [221] can identify biomarkers with consistent expression patterns in tumor progression, with potential prognostic and predictive implications in large patient populations. Our methodologies form a conceptually innovative framework encompassing various available information from research laboratories to healthcare systems for the discovery of biomarkers and therapeutic targets, including new and repositioning drugs, ultimately improving cancer patient survival outcomes.

## 8. Patents

The artificial intelligence methodology using Boolean implication networks based on prediction logic for drug discovery was filed under PCT/US22/75136.

## Figures and Tables

**Figure 1 cells-12-00101-f001:**
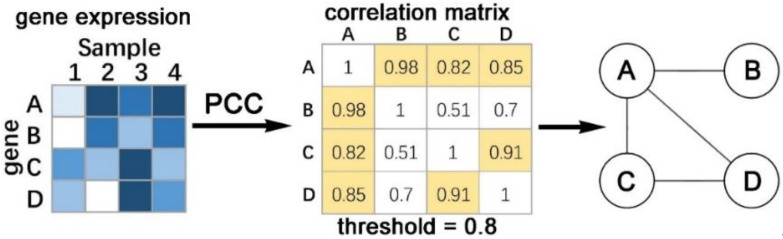
Constructing a relevance network using Pearson Correlation Coefficient (PCC).

**Figure 2 cells-12-00101-f002:**
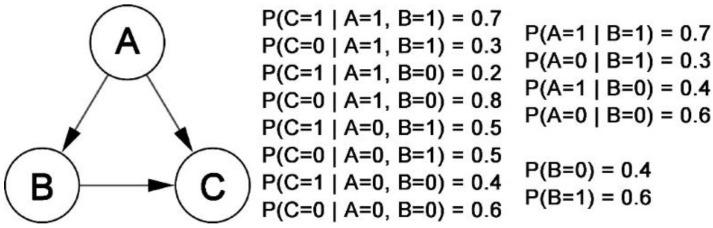
Example of a discrete Bayesian network.

**Figure 3 cells-12-00101-f003:**
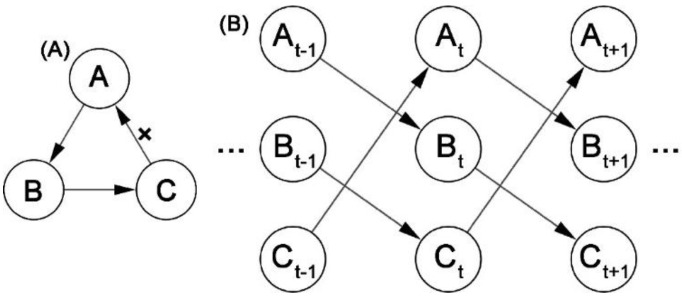
Example of a static Bayesian network (**A**) and a dynamic Bayesian network (**B**).

**Figure 4 cells-12-00101-f004:**
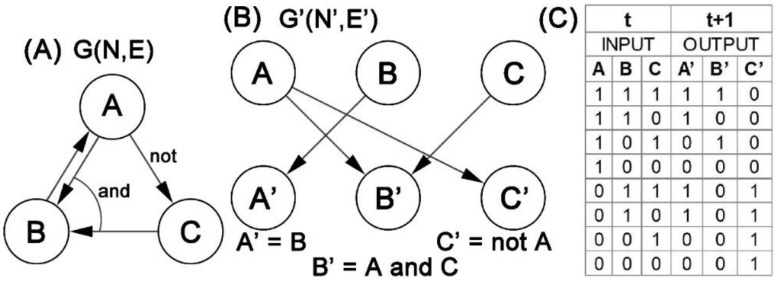
Example of a Boolean network. (**A**). A simple Boolean network $G\left(N,\;E\right)$. (**B**). The state transition corresponding to the linkage graph $G^{\prime }\left(N^{\prime },\;E^{\prime }\right)$ of A. (**C**). The truth table of this network.

**Figure 5 cells-12-00101-f005:**
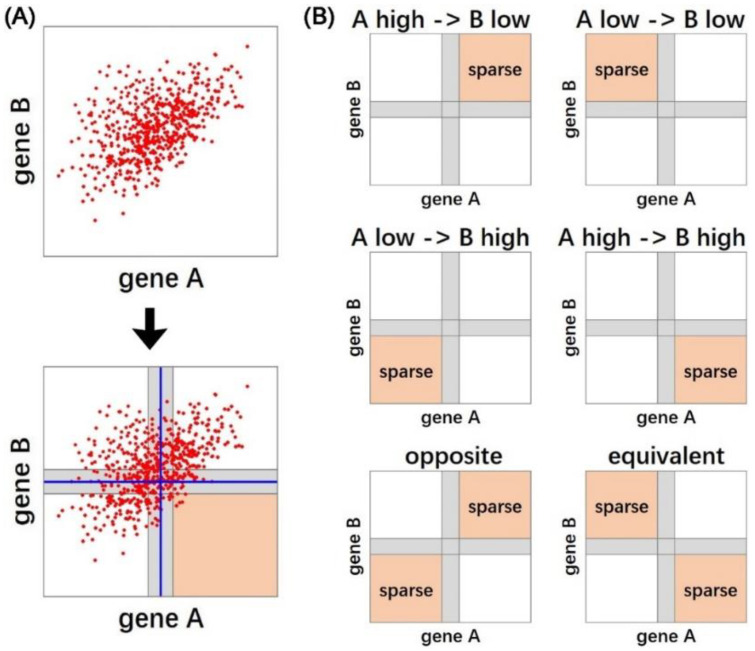
Example of a Boolean implication rule. (**A**). Inducing an implication rule based on quadrants of the categorized expression levels of gene A and gene B. The intermediate areas in gray are removed from the implication rule induction. (**B**). Six specific implication rules connecting gene A and gene B.

**Figure 6 cells-12-00101-f006:**
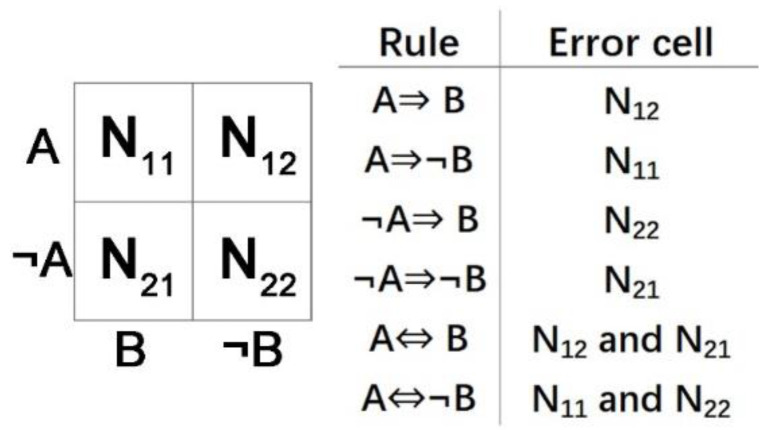
Contingency table of the Boolean implication rule and their corresponding error cells in prediction logic.

**Figure 7 cells-12-00101-f007:**
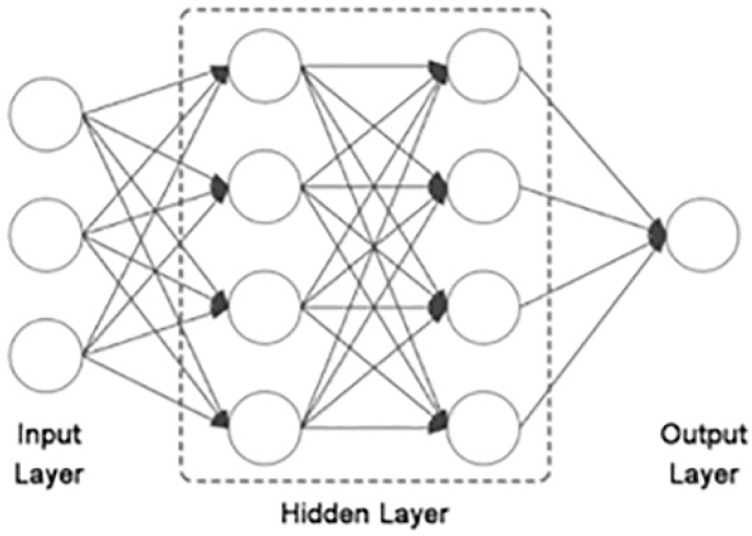
The basic structure of neural networks.

**Figure 8 cells-12-00101-f008:**
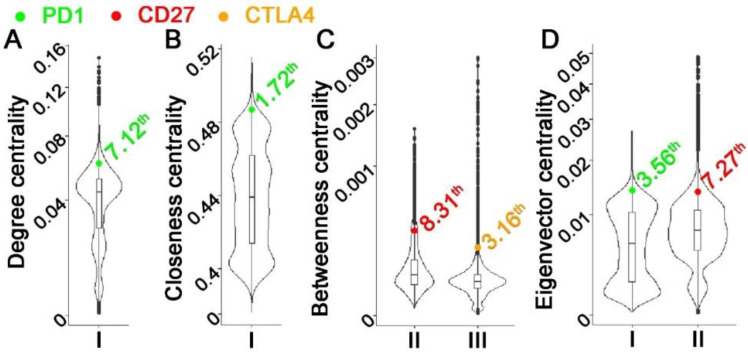
Distribution of centrality metrics in single-cell gene co-expression networks with *CD27, CTLA4,* or *PD1* ranked within the top 10th percentile. Each subplot represented a centrality metric: (**A**). Degree centrality; (**B**). Closeness centrality; (**C**). Betweenness centrality; (**D**). VoteRank centrality. Each violin plot showed the distribution of the centrality metric in one specific network: I. T-cell PBL gene co-expression network in normal samples. II. T-cell PBL gene co-expression network in NSCLC patients. III. T-cell gene co-expression network in NSCLC tumors.

**Figure 9 cells-12-00101-f009:**
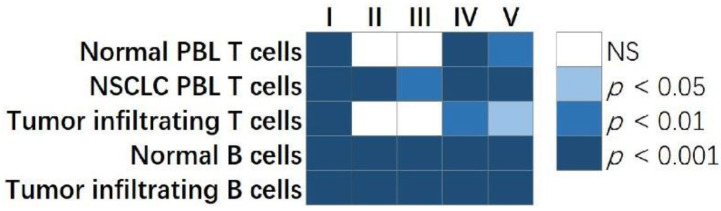
The comparison of centrality metrics of our published single B-cell network vs. randomly selected networks with the same number of genes. The *p* values showed the percentage of randomly selected genes having a higher ranked average centrality metric than the clinically relevant single B-cell network. Each column in the plot showed a centrality metric: I. Degree centrality; II. Eigenvector centrality; III. Closeness centrality; IV. Betweenness centrality; V. VoteRank centrality. Each row represented a single-cell gene co-expression network constructed in normal PBL T cells, NSCLC PBL T cells, tumor infiltrating T cells, normal B cells, and tumor infiltrating B cells, respectively. NS: not statistically significant.

**Table 1 cells-12-00101-t001:** Summary of software for multi-omics data processing, pathway analysis, and network inferencing in bulk tumors and single cells.

Purpose	Software
Data Processing	
Multi-omics data	GATK [43]
Copy number variation	PennCNV-Affy [51], CGHbase [52], CGHcall [53], GISTIC2.0 [54]
Single-cell RNA sequencing	Ginkgo [68], *STAR* aligner [72], SAMtools [73], DEsingle [75], scGNN [185]
Pathway Analysis	GSEA [77], ToppFun [78], Qiagen IPA, Adviata iPathwayGuide
Stromal and Immune Infiltration and Cell Activity	ESTIMATE [90], xCell [91], TIMER 2.0 [92,93,94], CIBERSORTx [95], MCP-counter [96]
Drug Discovery and Repositioning	CMap [26,27]
Network Inferencing Methods	GeNeCK [197]
Relevance networks	MiBiOmics [108], OmicsAnalyst [109], CorDiffViz [110]
Bayesian networks	CBNplot [144], TETRAD IV [147]
Boolean networks	SCNS [166]
PLBINs	Proprietary
Classification	Weka [198] (including neural networks and Bayesian networks)

**Table 2 cells-12-00101-t002:** Information of single-cell gene co-expression networks. The network nodes are genes and network edges are computed gene associations (one-tailed *z*-tests, *p* < 0.05, 95% confidence interval).

Patient Cohort	Network (Number of Cell Samples)	Number of Network Nodes	Number of Network Edges
GSE84789	NATs: B-cell gene co-expression (*n* = 96)	13,797	21,474,928
	Tumors: B-cell gene co-expression (*n* = 96)	13,420	6,298,276
GSE151531	Healthy donors: T-cell PBL gene co-expression (*n* = 431)	16,143	5,246,634
	NSCLC Patients: T-cell PBL gene co-expression (*n* = 92)	11,082	2,138,492
GSE151537	Tumors: T-cell gene co-expression (*n* = 2950)	20,171	7,805,674

**Table 3 cells-12-00101-t003:** Computational complexity and running time of each centrality method. *N*: the number of nodes (genes). *E*: the number of edges (gene associations).

Method	Complexity	Running Time (of a Network with 20 Million Edges)
PLBIN	$O\left(N^{2}\right)$	67 min
Degree Centrality	$O\left(N\right)$	0.02 s
Eigenvector Centrality	$O\left(N+E\right)$	89 s
Closeness Centrality	$O\left(NE\right)$	121 min
Betweenness Centrality	$O\left(N^{2}logN+NE\right)$	24 h
VoteRank Centrality	$O\left(E+rlogN+r\frac{E^{2}}{N^{2}}\right)$	53 h

**Table 4 cells-12-00101-t004:** Counts of concordant significant associations of each centrality metric with tumorigenesis, proliferation, and patient survival in the multi-omics networks.

Centrality Metric	Tumorigenesis (mRNA Expression)	Tumorigenesis (Protein Expression)	Proliferation (CRISPR-Cas9)	Proliferation (RNAi)	Patient Survival	Sum
Degree Centrality	3	2	3	3	2	13
Eigenvector Centrality	4	1	5	5	3	18
Closeness Centrality	4	1	5	4	2	16
Betweenness Centrality	0	1	2	2	1	6
VoteRank Centrality	0	0	4	3	1	8
**Sum**	11	5	19	17	9	61

## Data Availability

The data was published and publicly available in the cited manuscripts. An earlier implementation of our Prediction Logic Boolean Implication Networks (PLBINs) was released in SourceForge: https://sourceforge.net/projects/genet-cnv/ (accessed on 21 December 2022). The current version of the software is patented for the discovery of biomarkers and therapeutic targets and is not released.

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
