# Peer review of "Inferencing Bulk Tumor and Single-Cell Multi-Omics Regulatory Networks for Discovery of Biomarkers and Therapeutic Targets"

_cells, 2022, doi:10.3390/cells12010101_

Round 1

Reviewer 1 Report

In this manuscript, Ye Q et al. summarize methods for multi-omics data harmonization and inferencing molecular networks to discover biomarkers and therapeutic targets. This review comprehensively presents a general description of related content. However, there are a few points that need to be addressed:

1.In Section 3.3, several algorithms can be used to estimate the abundance of immune cells in tumor bulk, not mentioned here. Take the "MCPcounter" as an example.

2. In this review, no comparative data were presented. It's hard to conclude that the Boolean implication networks the author mentioned compasses other methods in constructing multi-omics network.

Author Response

Reviewer 1

Comments and Suggestions for Authors

Reviewer: In this manuscript, Ye Q et al. summarize methods for multi-omics data harmonization and inferencing molecular networks to discover biomarkers and therapeutic targets. This review comprehensively presents a general description of related content. However, there are a few points that need to be addressed:

Authors: We thank the reviewer for all constructive comments.

Reviewer: 1.In Section 3.3, several algorithms can be used to estimate the abundance of immune cells in tumor bulk, not mentioned here. Take the "MCPcounter" as an example.

Authors: We thank the reviewer for the comment. MCP-counter is now added in Section 3.3.

Reviewer: 2. In this review, no comparative data were presented. It's hard to conclude that the Boolean implication networks the author mentioned compasses other methods in constructing multi-omics network.

Authors: We appreciate this comment. The comparative data of our Prediction Logic Boolean Implication Networks (PLBINs) and other methods were published previously [1,2]. It is now added to the manuscript (Section 4.9).

The precision and false discovery rate (FDR) of the gene coexpression networks were evaluated as follows [1]. The biological relevance of computationally derived coexpression relations was evaluated with five gene set collections (positional, curated, motif, computational, and Gene Oncology) and canonical pathway databases from the MSigDB [3]. A coexpression relation was labeled as a true positive (TP) if the pair of genes belonged to the same gene set or pathway in any investigated database. If a pair of genes did not share any gene set or pathway, the coexpression relation was labeled a false positive (FP). A coexpression relation was considered non-discriminatory (ND) if at least one gene in the pair was not annotated in a database [4]. Coexpression relations labeled as ND were excluded in the evaluation as they were not confirmed. With precision defined as TP/(TP + FP), the precision of our identified smoking-mediated coexpression networks in NSCLC patient tumors was 100% [1]. The null distribution was generated in 1,000 random permutations of the class labels in the test cohort. The precision of the smoking-mediated coexpression networks was significant at p<0.001, with no TP generated in the random tests. With FDR defined as the average of FP/(TP+FP) in 1,000 permutations, the FDR of the smoking-mediated coexpression networks in NSCLC patient tumors was 0.0099. In contrast, gene association networks based on Pearson’s correlation coefficients were not able to identify any gene coexpression relations using the same methodology on the same datasets [1]. In the evaluation of our identified 21 NSCLC prognostic gene signatures [5] using the NCI Director’s Challenge Study [6],  the gene coexpression relations derived from the training cohort using our PLBINs could be successfully reproduced in both test cohorts with significantly high precision (precision = 1 for 18 out of 21 gene signatures) and low FDR (FDR < 0.1) [2]. As a comparison, Bayesian belief networks (BBNs) implemented in TETRAD IV [7] did not identify any coexpression relations validated in all three cohorts [2].  Other Boolean implication networks by Sahoo et al [8] did not identify coexpression with many of the major NSCLC hallmarks, making it infeasible to select marker genes with crosstalk with multiple signaling pathways. Our PLBIN can tune preset parameters to increase precision and reduce FDR. Other Boolean implication networks do not provide further information on tuning the parameters. In the genome-scale evaluation, our PLBIN achieved significantly high precision in 1,000 random tests (p < 0.05), whereas the precision of other Boolean implication networks by Sahoo et al [8] was not significant in 1,000 random tests (p =0.21).

References

  1. Guo, N.L.; Wan, Y.W. Pathway-based identification of a smoking associated 6-gene signature predictive of lung cancer risk and survival. Artif. Intell. Med 2012.
  2. Guo, N.L.; Wan, Y.W. Network-based identification of biomarkers coexpressed with multiple pathways. Cancer Inform 2014, 13, 37-47.
  3. Subramanian, A.; Tamayo, P.; Mootha, V.K.; Mukherjee, S.; Ebert, B.L.; Gillette, M.A.; Paulovich, A.; Pomeroy, S.L.; Golub, T.R.; Lander, E.S.; et al. Gene set enrichment analysis: A knowledge-based approach for interpreting genome-wide expression profiles. Proceedings of the National Academy of Sciences of the United States of America 2005, 102, 15545-15550.
  4. Ucar, D.; Neuhaus, I.; Ross-MacDonald, P.; Tilford, C.; Parthasarathy, S.; Siemers, N.; Ji, R.R. Construction of a reference gene association network from multiple profiling data: application to data analysis. Bioinformatics (Oxford, England) 2007, 23, 2716-2724.
  5. Wan, Y.W.; Beer, D.G.; Guo, N.L. Signaling pathway-based identification of extensive prognostic gene signatures for lung adenocarcinoma. Lung cancer (Amsterdam, Netherlands) 2012, 76, 98-105.
  6. Shedden, K.; Taylor, J.M.; Enkemann, S.A.; Tsao, M.S.; Yeatman, T.J.; Gerald, W.L.; Eschrich, S.; Jurisica, I.; Giordano, T.J.; Misek, D.E.; et al. Gene expression-based survival prediction in lung adenocarcinoma: a multi-site, blinded validation study. Nat. Med 2008, 14, 822-827.
  7. Taylor, I.W.; Linding, R.; Warde-Farley, D.; Liu, Y.; Pesquita, C.; Faria, D.; Bull, S.; Pawson, T.; Morris, Q.; Wrana, J.L. Dynamic modularity in protein interaction networks predicts breast cancer outcome. Nature biotechnology 2009, 27, 199-204.
  8. Sahoo, D.; Dill, D.L.; Gentles, A.J.; Tibshirani, R.; Plevritis, S.K. Boolean implication networks derived from large scale, whole genome microarray datasets. Genome biology 2008, 9, R157.

Reviewer 2 Report

Qing Ye and Nancy Lan Guo aimed to integrate multi-omics profiles in a patient cohort with large-scale patient EMRs such as the SEER-Medicare cancer registry combined with extensive external validation can identify potential biomarkers applicable in large patient populations. In their study, multi-omics data harmonization methods were introduced, and common approaches to molecular network inference were summarized. The study was well designed and performed, but there are still several comments for the authors.

Major:

1. The authors have demonstrated that “Our Boolean implication networks based on prediction logic have advantages over other methods in constructing genome-scale multi-omics networks in bulk tumors and single cells in terms of computational efficiency, scalability, and accuracy”. Is there any developed webtools for practical application, such as shiny?

2. Was the SEER-Medicare data accessed by the authors with special permission from the official organization (https://seer.cancer.gov/)? This is a special part of the SEER database that need additional applications from the authors.

3. The author have listed lots of formulas or illustrations of algorithms in this review. However, it seems they did not summarize all these methods in a table or figure which was commonly used in reviews.

4. The authors demonstrated that their Boolean implication networks were better than other methods. But it seems that they did not provide robust evidence in this manuscript. If so, please discuss and comment.

Minor:

1. The authors should further recheck whether the structure of this manuscript met the criteria of MDPI review.

2. In the Patents part, the authors claimed a patent of Boolean implication networks. Was the patent individual? Did it belong to any institutions? If so, please modify the Conflicts of Interest part of the manuscript.

Author Response

Reviewer 2

Comments and Suggestions for Authors

Reviewer: Qing Ye and Nancy Lan Guo aimed to integrate multi-omics profiles in a patient cohort with large-scale patient EMRs such as the SEER-Medicare cancer registry combined with extensive external validation can identify potential biomarkers applicable in large patient populations. In their study, multi-omics data harmonization methods were introduced, and common approaches to molecular network inference were summarized. The study was well designed and performed, but there are still several comments for the authors.

Authors: We thank the reviewer for all constructive comments.

Major:

Reviewer: 1. The authors have demonstrated that “Our Boolean implication networks based on prediction logic have advantages over other methods in constructing genome-scale multi-omics networks in bulk tumors and single cells in terms of computational efficiency, scalability, and accuracy”. Is there any developed webtools for practical application, such as shiny?

Authors: An earlier implementation of our Prediction Logic Boolean Implication Networks (PLBINs) was released in SourceForge: https://sourceforge.net/projects/genet-cnv/.  The current version of the software is patented for the discovery of biomarkers and therapeutic targets and is not released.

Reviewer: 2. Was the SEER-Medicare data accessed by the authors with special permission from the official organization (https://seer.cancer.gov/)? This is a special part of the SEER database that need additional applications from the authors.

Authors: The SEER-Medicare data license was purchased from the NCI. We have permission to use the data and our publications were approved by the official organization.

Reviewer: 3. The author have listed lots of formulas or illustrations of algorithms in this review. However, it seems they did not summarize all these methods in a table or figure which was commonly used in reviews.

Authors: A summary table is now added in Section 4.9.

Reviewer: 4. The authors demonstrated that their Boolean implication networks were better than other methods. But it seems that they did not provide robust evidence in this manuscript. If so, please discuss and comment.

Authors: We appreciate this comment. The comparative data of our Prediction Logic Boolean Implication Networks (PLBINs) and other methods were published previously [1,2]. It is now added to the manuscript (Section 4.9).

The precision and false discovery rate (FDR) of the gene coexpression networks were evaluated as follows [1]. The biological relevance of computationally derived coexpression relations was evaluated with five gene set collections (positional, curated, motif, computational, and Gene Oncology) and canonical pathway databases from the MSigDB [3]. A coexpression relation was labeled as a true positive (TP) if the pair of genes belonged to the same gene set or pathway in any investigated database. If a pair of genes did not share any gene set or pathway, the coexpression relation was labeled a false positive (FP). A coexpression relation was considered non-discriminatory (ND) if at least one gene in the pair was not annotated in a database [4]. Coexpression relations labeled as ND were excluded in the evaluation as they were not confirmed. With precision defined as TP/(TP + FP), the precision of our identified smoking-mediated coexpression networks in NSCLC patient tumors was 100% [1]. The null distribution was generated in 1,000 random permutations of the class labels in the test cohort. The precision of the smoking-mediated coexpression networks was significant at p<0.001, with no TP generated in the random tests. With FDR defined as the average of FP/(TP+FP) in 1,000 permutations, the FDR of the smoking-mediated coexpression networks in NSCLC patient tumors was 0.0099. In contrast, gene association networks based on Pearson’s correlation coefficients were not able to identify any gene coexpression relations using the same methodology on the same datasets [1]. In the evaluation of our identified 21 NSCLC prognostic gene signatures [5] using the NCI Director’s Challenge Study [6],  the gene coexpression relations derived from the training cohort using our PLBINs could be successfully reproduced in both test cohorts with significantly high precision (precision = 1 for 18 out of 21 gene signatures) and low FDR (FDR < 0.1) [2]. As a comparison, Bayesian belief networks (BBNs) implemented in TETRAD IV [7] did not identify any coexpression relations validated in all three cohorts [2].  Other Boolean implication networks by Sahoo et al [8] did not identify coexpression with many of the major NSCLC hallmarks, making it infeasible to select marker genes with crosstalk with multiple signaling pathways. Our PLBIN can tune preset parameters to increase precision and reduce FDR. Other Boolean implication networks do not provide further information on tuning the parameters. In the genome-scale evaluation, our PLBIN achieved significantly high precision in 1,000 random tests (p < 0.05), whereas the precision of other Boolean implication networks by Sahoo et al [8] was not significant in 1,000 random tests (p =0.21).

Minor:

Reviewer: 1. The authors should further recheck whether the structure of this manuscript met the criteria of MDPI review.

Authors: We confirmed the structure of the manuscript meets the journal requirements for a review.

Reviewer: 2. In the Patents part, the authors claimed a patent of Boolean implication networks. Was the patent individual? Did it belong to any institutions? If so, please modify the Conflicts of Interest part of the manuscript.

Authors: The patent inventor is Dr. Nancy Guo. The patent was filed and owned by West Virginia University. The Conflicts of Interest is modified. 

References

  1. Guo, N.L.; Wan, Y.W. Pathway-based identification of a smoking associated 6-gene signature predictive of lung cancer risk and survival. Artif. Intell. Med 2012.
  2. Guo, N.L.; Wan, Y.W. Network-based identification of biomarkers coexpressed with multiple pathways. Cancer Inform 2014, 13, 37-47.
  3. Subramanian, A.; Tamayo, P.; Mootha, V.K.; Mukherjee, S.; Ebert, B.L.; Gillette, M.A.; Paulovich, A.; Pomeroy, S.L.; Golub, T.R.; Lander, E.S.; et al. Gene set enrichment analysis: A knowledge-based approach for interpreting genome-wide expression profiles. Proceedings of the National Academy of Sciences of the United States of America 2005, 102, 15545-15550.
  4. Ucar, D.; Neuhaus, I.; Ross-MacDonald, P.; Tilford, C.; Parthasarathy, S.; Siemers, N.; Ji, R.R. Construction of a reference gene association network from multiple profiling data: application to data analysis. Bioinformatics (Oxford, England) 2007, 23, 2716-2724.
  5. Wan, Y.W.; Beer, D.G.; Guo, N.L. Signaling pathway-based identification of extensive prognostic gene signatures for lung adenocarcinoma. Lung cancer (Amsterdam, Netherlands) 2012, 76, 98-105.
  6. Shedden, K.; Taylor, J.M.; Enkemann, S.A.; Tsao, M.S.; Yeatman, T.J.; Gerald, W.L.; Eschrich, S.; Jurisica, I.; Giordano, T.J.; Misek, D.E.; et al. Gene expression-based survival prediction in lung adenocarcinoma: a multi-site, blinded validation study. Nat. Med 2008, 14, 822-827.
  7. Taylor, I.W.; Linding, R.; Warde-Farley, D.; Liu, Y.; Pesquita, C.; Faria, D.; Bull, S.; Pawson, T.; Morris, Q.; Wrana, J.L. Dynamic modularity in protein interaction networks predicts breast cancer outcome. Nature biotechnology 2009, 27, 199-204.
  8. Sahoo, D.; Dill, D.L.; Gentles, A.J.; Tibshirani, R.; Plevritis, S.K. Boolean implication networks derived from large scale, whole genome microarray datasets. Genome biology 2008, 9, R157.

Round 2

Reviewer 1 Report

Ye Q et al. have responded satisfactorily to the comments.